# Timing and Amplitude of Light Exposure, Not Photoperiod, Predict Blood Lipids in Arctic Residents: A Circadian Light Hypothesis

**DOI:** 10.3390/biology14070799

**Published:** 2025-07-01

**Authors:** Denis Gubin, Sergey Kolomeichuk, Konstantin Danilenko, Oliver Stefani, Alexander Markov, Ivan Petrov, Kirill Voronin, Marina Mezhakova, Mikhail Borisenkov, Aislu Shigabaeva, Julia Boldyreva, Julianna Petrova, Larisa Alkhimova, Dietmar Weinert, Germaine Cornelissen

**Affiliations:** 1Department of Biology, Tyumen Medical University, 625023 Tyumen, Russia; 2Laboratory for Chronobiology and Chronomedicine, Research Institute of Biomedicine and Biomedical Technologies, Tyumen Medical University, 625023 Tyumen, Russia; kvdani@mail.ru (K.D.); h_aislu@mail.ru (A.S.); tgma.06@mail.ru (J.B.); 3Tyumen Cardiology Research Centre, Tomsk National Research Medical Center, Russian Academy of Science, 119991 Tyumen, Russia; 4Laboratory for Genomics, Proteomics, and Metabolomics, Research Institute of Biomedicine and Biomedical Technologies, Medical University, 625023 Tyumen, Russia; sergey_kolomeychuk@rambler.ru (S.K.); alexdoktor@inbox.ru (A.M.); lirik92@list.ru (K.V.); chiz.maslova@yandex.ru (M.M.); 5Laboratory of Genetics, Institute of Biology of the Karelian Science Center of the Russian Academy of Sciences, 185910 Petrozavodsk, Russia; 6Institute of Neurosciences and Medicine, 630117 Novosibirsk, Russia; 7Department Engineering and Architecture, Lucerne University of Applied Sciences and Arts, 6048 Horw, Switzerland; oliver.stefani@hslu.ch; 8Department of Biological & Medical Physics UNESCO, Medical University, 625023 Tyumen, Russia; petrovtokb@mail.ru (I.P.); pimtmn@mail.ru (J.P.); 9Department of Molecular Immunology and Biotechnology, Institute of Physiology of the Federal Research Centre Komi Science Centre of the Ural Branch of the Russian Academy of Sciences, 167982 Syktyvkar, Russia; borisenkov@physiol.komisc.ru; 10School of Natural Sciences, University of Tyumen, 15a Perekopskaya St., 625003 Tyumen, Russia; l.e.alkhim@gmail.com; 11Institute of Biology/Zoology, Martin Luther University, Halle-Wittenberg, 06120 Halle, Germany; dietmar.weinert@zoologie.uni-halle.de; 12Department of Integrated Biology and Physiology, University of Minnesota, Minneapolis, MN 55455, USA; corne001@umn.edu

**Keywords:** light exposure, circadian rhythm, arctic, light dynamic range, phase, blood lipids, photoperiod, high-density lipoprotein cholesterol, low-density lipoprotein cholesterol, triglycerides

## Abstract

Arctic residents exhibit seasonal lipid variations in response to changes in light exposure (LE). This study examined how LE features (e.g., day length (photoperiod duration), timing, and dynamic range) predict lipid profiles. Actigraphy and blood samples from 27 residents across seasons and their melatonin profiles (n = 13) were analyzed. LE timing and dynamic range and melatonin acrophase significantly predicted lipid levels, which were independent of the photoperiod. Earlier LE was associated with higher HDL-C while elevated nighttime blue light was linked to higher TC, LDL-C, and TG. A greater LE dynamic range correlated with higher HDL-C and lower TG/HDL ratio, and an earlier melatonin acrophase predicted lower TG/HDL. These results suggest that LE timing and dynamic range influence lipid profiles in high-latitude residents, highlighting the potential for interventions to modulate cardiovascular risk.

## 1. Introduction

An intimate connection exists between lipid metabolism and the circadian system, with disruptions in circadian rhythms frequently associated with altered blood lipid profiles [1,2,3,4,5,6,7,8,9,10,11,12,13,14,15,16]. Light is a major synchronizer of circadian rhythms, including those of metabolic processes. The connection between light signaling and lipid metabolism is supported by evidence from both animal models [8,9] and human studies [2,10,16]. For instance, exposure to dim light at night disrupts lipid metabolic homeostasis in rats, leading to increased lipid storage in the liver, altered adipokine levels, and deregulation of key metabolic transcription factors, suggesting a potential mechanism linking even a dim light at night to metabolic diseases [9]. In humans, even low blue light exposure between 9:30 p.m. and 0:30 a.m. was associated with a higher BMI, especially in individuals with a specific genetic variant of a melatonin receptor [11]. A recent nationwide study of over 10,000 Chinese adults aged 45 years and older found that greater long-term exposure to outdoor light at night (LAN) was significantly associated with an increased prevalence of dyslipidemia, including high LDL-cholesterol (LDL-C), high triglycerides (TG), and low HDL-cholesterol (HDL-C) [16]. Previously, a time-dependent association between retinal ganglion cell (RGC) loss in glaucoma patients and adverse changes in lipid metabolism, particularly in morning HDL-C and TG, and a shift in the morning–evening gradient of LDL-C correlated with RGC damage and specific CLOCK gene polymorphisms [10]. The study by Kent et al. [13] defined phase–response curves (PRCs) for lipids and hepatic proteins in humans, demonstrating that combined light and food stimuli predictably alter their circadian rhythms, with phase shifts dependent on the stimulus timing and distinct from those observed for melatonin, though TG was closest to melatonin.

Recommendations for light hygiene [17] are rarely met in urban residents, contributing to various negative health outcomes, including metabolic issues [11,18,19,20,21,22]. Light hygiene is compromised even more dramatically at high latitudes, where natural light exposure varies drastically depending on seasons. Until recently, longitudinal field research on objectively measured circadian physiology, sleep, and light exposure during different seasons in real-world Arctic settings was lacking. We addressed this gap by conducting a longitudinal study [11,12] of community-dwelling Arctic residents using wearable light-measuring data loggers [23,24] to capture seasonal variations. Due to the rapid changes in day length around the solstices at high latitudes, we used strictly fixed dates for our field experiments, as close as possible to the winter solstice (natural photoperiod duration of less than 1 h), spring equinox (photoperiod duration is about 12 h), and summer solstice (natural photoperiod is longer than 23 h). Our Arctic study demonstrated that changes in light exposure (LE) and blue light exposure (BLE) patterns during winter and summer solstices (WS and SS) compared to spring equinox (SE), characterized by delayed blue light and melatonin rhythms, were associated with adverse changes in TG, HDL-C, and cortisol. These results highlighted the importance of optimal light hygiene for maintaining metabolic health at high latitudes [12]. Three key features characterize seasonal variations in LE and BLE among Arctic residents: (1) substantial alterations in photoperiod duration/day-length, (2) significant shifts in the phase of light exposure, including the timing of maximal and minimal light exposure, and (3) considerable changes in the dynamic range of light exposure. A novel index of circadian light hygiene consists of the normalized amplitude of light/blue light exposure (NA BLE) [19,24,25]. Extreme changes in photoperiod in the Arctic may lead to differential circadian entrainment strategies: significantly stronger coupling between BLE and physical activity acrophases in non-native (NN) residents compared to native (N) residents was found, with N residents exhibiting such a coupling primarily during the winter solstice when light exposure is low [26]. Solstice-related BLE phase delays also correlated with delays in physical activity only in NN residents, highlighting potential differences in circadian adaptation. Importantly, individual variability within seasons further modulates the above-mentioned three seasonal aspects of LE and may be associated with individual lipid profiles.

To investigate which of these factors exerts the most substantial influence on lipid profiles and their seasonal fluctuations, we employed a multivariate analysis of covariance to control for potential confounding variables, such as age, sex, and indigenous status.

## 2. Materials and Methods

### 2.1. Study Inclusion and Exclusion Criteria

Data for this study were collected as part of the first phase of the “Light Arctic” project, which targeted residential populations in three locations in Russia: Salekhard (66°53′ N, 66°60′ E), Aksarka settlement (66°33′ N, 67°48′ E), and Urengoy town (65°58′ N, 76°63′ E). Of the 156 volunteers initially enrolled (age range: 12–59 years), 62 participants provided complete datasets meeting the study’s quality standards, as outlined in prior publications [11,12]. A subsample of this study comprised 27 participants (75% women, 19 NN, 8 N) who provided data across three seasons (winter and summer solstices and spring equinox). At the completion of actigraphy, weight and height measurements were obtained at local healthcare facilities using calibrated equipment. Body mass index (BMI) was calculated as weight (in kilograms) divided by the square of height (in meters).

### 2.2. Actigraphy

Utilizing ActTrust 2 devices (Condor Instruments, São Paulo, Brazil), participants’ activity and light exposure were tracked for seven days, building on established methods [11,12]. The devices recorded light intensity (lux) and its spectral components (infrared, red, green, blue, UVA, UVB, μw/cm^2^) every minute. The ActStudio software 1.0.25 yielded estimates of parametric (MESOR, 24 h amplitude and acrophase for LE and BLE) and non-parametric (BLE M10 and L5 and their onsets) endpoints. To rigorously assess the amplitude of BLE and to reduce bias from profound inter-individual variability, the Normalized Amplitude of BLE (NA BLE) was calculated as the ratio of the BLE 24 h amplitude to the corresponding 24 h MESOR. LE and BLE were assessed in a natural setting, reflecting participants’ real-world light exposure to inform the human-centric lighting intervention in the ongoing second phase of the study.

### 2.3. Blood Lipids

Following the conclusion of actigraphy monitoring, participants presented at the clinic in the morning (between 8:00 and 9:00 a.m.), after a 12 h period of fasting, for blood sampling. Participants were encouraged to maintain their usual daily activities and dietary habits. Samples were obtained via venipuncture of the ulnar vein using vacutainers. Collected blood samples were processed using a Tecan HydroFlex microplate mixer (Grödig, Austria). Plasma lipid profiles, including total cholesterol (TC), high-density lipoprotein cholesterol (HDL-C), low-density lipoprotein cholesterol (LDL-C), and triglycerides (TG), were assessed photometrically with a Mindray BS-380 automatic analyzer (Mindray, Shenzhen, China). All biochemical analyses were performed within the accredited laboratory of the Research Institute of Biomedicine and Biomedical Technologies at Tyumen Medical University, as described in detail elsewhere [12].

### 2.4. Melatonin

Salivary melatonin was determined using a protocol described in detail in our previous work [12]. Thirteen participants each provided 21 saliva samples in each season across a 24 h period on their final day of actigraphy, adhering to personalized schedules created from their sleep diary records. Sampling was most frequent before and during sleep, with additional samples collected throughout the daytime hours. To control for light’s effect on melatonin, participants were instructed to wear efficacy-tested blue-blocking glasses (Surgut Super O88, Surgut, Russia) and avoid artificial light exposure, following specific saliva collection guidelines (e.g., after rinsing the mouth and a 10-minute wait). Samples were analyzed using the Novolytix Laboratories, Witterswil, Switzerland saliva melatonin ELISA kit, which has demonstrated validity and comparability to RIA, as shown by Burgess et al. [27] and confirmed in our prior study [12]. Raw melatonin data underwent quality control procedures to remove invalid values, such as those resulting from empty collection tubes, extended gaps in sampling, or extreme outliers (defined as changes > 2 SDs between consecutive samples), using criteria detailed in [12].

### 2.5. Chronotype

All participants completed the Horne–Ostberg Morningness–Eveningness Questionnaire (MEQ) [28]. However, due to a restricted range of MEQ scores among the subset of participants who met the study’s criteria for both actigraphy and melatonin data collection in each season (n = 13; MEQ range: 45–64, mean ± SD: 51.6 ± 5.1), the sample was not representative of distinct morning or evening chronotypes, as previously described elsewhere [12].

### 2.6. Data and Statistical Analyses

Statistical analyses were performed using Libre Office Calc (Berlin, Germany) and STATISTICA 64 software (Palo Alto, CA, USA). Participants provided data, including a week of actigraphy monitoring and a single blood sample collected between 8:00 and 9:00 a.m. To determine the appropriate statistical tests, the normality of each variable’s distribution was examined using the Shapiro–Wilk and Kolmogorov–Smirnov tests. Depending on the results of those tests, Student’s *t*-tests, Mann–Whitney U tests, or Kruskal–Wallis tests were employed to compare blood lipid values and actigraphy measures across seasons, population groups, and sex. Linear regression was used to assess the relationship between blood lipids and actigraphy measurements, with the Benjamini–Hochberg false discovery rate (FDR = 0.1) employed to correct for multiple testing. Variance Inflation Factors (VIF) was assessed to check for multi-collinearity between variables in the models. To identify significant predictors of blood lipids while controlling for potential confounders, multivariable ANCOVA was performed, adjusting for photoperiod duration (approximated as 1 h during WS, 12 h during SE, and 23 h during SS), age, sex, and indigeneity. Due to multicollinearity with age, sex, lipid values, and amplitude-phase light exposure measures, BMI was not used in the models predicting lipid levels. A *p*-value of less than 0.05 was considered to indicate statistical significance.

## 3. Results

Table 1 presents correlation coefficients between blood lipids and LE measures in Arctic residents. Photoperiod duration and general measures of light quantity, such as 24 h and daytime mean LE and BLE (MESOR, M10, and Amplitude), were not significantly associated with any blood lipid parameter. However, significant associations, surviving Benjamini–Hochberg correction (FDR = 0.1), were observed between light exposure phase (LE Acrophase, BLE Acrophase) and HDL-C, and between the dynamic range of light exposure (NA BLE) and TG and the TG/HDL ratio. Furthermore, some L5 BLE results were significantly associated with lipids. Commonly, high HDL-C is considered beneficial for its role in removing cholesterol from arteries, while elevated TC, LDL-C, and TG are associated with an increased risk of cardiovascular disease [29,30].

Table 2 presents a summary of results from multiple regression analyses examining the relationships between single light exposure metrics, including daytime, nocturnal, and 24 h light quantity, timing, and dynamic range, and blood lipids, after adjusting for photoperiod duration. Consistent with previous findings, photoperiod duration was not associated with any of the blood lipid parameters. Moreover, neither the 24 h average LE nor mean daytime LE was related to changes in blood lipid profiles among Arctic residents. However, an earlier timing of LE and a more robust dynamic range of LE as assessed by NA BLE were associated with higher HDL-C. Furthermore, higher TC, LDL-C, and TG were all associated with greater BLE at night, measured as L5, and a later onset of L5. Elevated TG and TG/HDL ratio exhibited the strongest associations with compromised circadian light hygiene, gauged by a smaller NA BLE.

After correcting for sex, age, and indigeneity in addition to photoperiod duration, an earlier acrophase (β = −0.257, partial η^2^ = 0.088, *p* = 0.009) and M10 onset of BLE (β = −0.218, partial η^2^ = 0.060, *p* = 0.033) predicted a higher HDL-C, suggesting benefits of earlier daylight exposure (Figure 1). On the contrary, a higher nighttime L5 of BLE predicted a higher TC (β = 0.290, partial η^2^ = 0.096, *p* = 0.006), higher LDL-C (β = 0.253, partial η^2^ = 0.070, *p* = 0.021), and higher TG (β = 0.221, partial η^2^ = 0.055, *p* = 0.039), while a later L5 onset of BLE predicted a higher TG (β = 0.246, partial η^2^ = 0.071, *p* = 0.019), illustrating hazards of nocturnal light. On the other hand, a larger NA BLE was linked to a higher HDL-C (β = 0.241, partial η^2^ = 0.063, *p* = 0.020) and lower TG (β = −0.304, partial η^2^ = 0.096, *p* = 0.006) and TG/HDL ratio (β = −0.390, partial η^2^ = 0.165, *p* < 0.001), suggesting benefits of proper circadian light hygiene on blood lipids. Interestingly, incorporating demographic co-factors attenuated the prediction of HDL-C by the acrophase and M10 Onset of BLE. However, these same co-factors improved the prediction of HDL-C using NA BLE, a measure of the dynamic range of BLE. Specifically, the addition of demographic co-factors increased the strength of the association (β) and explained variance (partial η^2^) for HDL-C prediction by NA BLE from −0.197/0.037 (photoperiod-adjusted model) to −0.241/0.063 (fully adjusted model). Average values for actigraphy metrics of LE/BLE for each separate season are provided in Table 3 and were previously discussed elsewhere [12].

While the mean and 24 h amplitude of salivary melatonin showed no association with blood lipids, the timing of the melatonin acrophase (peak) was a significant predictor of TG and TG/HDL. Specifically, an earlier melatonin acrophase correlated with lower TG (r = 0.454, *p* = 0.006) and TG/HDL (r = 0.510, *p* = 0.002), and borderline significantly with higher HDL-C (r = −0.282, *p* = 0.064), Table 1. These correlations were retained after adjusting for photoperiod, being strongest for TG and TG/HDL-C, as seen in Table 2. Notably, even after adjusting for sex, age, population, and photoperiod duration, an earlier melatonin acrophase remained the strongest independent predictor of a lower TG (β = 0.451, *p* = 0.026) and a lower TG/HDL-C (β = 0.464, *p* = 0.007).

A detailed examination of the temporal relationship between BLE and lipid metabolism was undertaken using the analytical workflow depicted in Figure 2. It allowed for the characterization of time-dependent associations between BLE and two lipids that were predicted by NA BLE: TG and HDL-C. Linear regression analysis demonstrated significant time-dependent correlations between BLE and both HDL-C and TG, revealing distinct diurnal patterns. Subsequent cosinor analysis confirmed significant circadian rhythmicity for both HDL-C (peak at 08:55) and TG (peak at 23:38).

Multi-factorial analyses also showed that TC increased with age (β = 0.261, *p* = 0.023), that HDL-C was lower in men than in women (β = −0.412, *p* < 0.001), and in NN than in N residents (β = −0.219, *p* = 0.031). They also showed that TG was higher in NN than in N residents (β = 0.333, *p* = 0.003), and that TG/HDL was higher in men than in women (β = 0.238, *p* = 0.044) and higher in NN than in N residents (β = 0.288, *p* = 0.010), Table 4.

## 4. Discussion

High-latitude environments are characterized by three key features of LE/BLE. First, the photoperiod duration exhibits extreme seasonal variation, from near-continuous daylight during SS to minimal daylight during WS and equal light and dark phases during the equinoxes (e.g., SE). Second, the timing of LE/BLE shifts with the seasons, being delayed during both WS and SS compared to SE [12]. Third, the dynamic range of LE/BLE varies with the seasons, as best quantified by NA BLE to account for inter-individual differences in average LE [19,24,25]. Previous work [12] linked these light environment changes to phase-amplitude shifts in melatonin and metabolic health decline. Consequently, phase-amplitude changes in LE/BLE, rather than the photoperiod duration, may be key predictors of blood lipid concentrations. This study of Arctic residents aimed at determining whether the timing and dynamic range of LE/BLE, rather than photoperiod, were key predictors of blood lipids.

In our multivariable analysis, controlling for age, sex, and native status, we found that the photoperiod duration did not significantly correlate with any lipid outcome. Instead, the timing of LE emerged as a crucial factor influencing TC and LDL-C, while NA BLE was the primary predictor for TG, HDL-C, and the TG/HDL-C ratio. These results underscore the importance of circadian light hygiene over mere photoperiod duration, suggesting that the timing and dynamic range of LE are more critical determinants of lipid profiles. The study also revealed that higher nocturnal BLE, L5, later onset of L5, and smaller NA BLE were associated with poorer lipid metabolism. Higher nighttime BLE predicted higher TC, LDL-C, and TG concentrations, while a smaller NA BLE was linked to elevated TG and TG/HDL ratio. These findings have direct clinical implications, as they establish a clear connection between specific LE patterns and adverse health outcomes, particularly in vulnerable Arctic populations. NA BLE, an index of circadian hygiene, is a critical metric for assessing the stability and regularity of an individual’s BLE. By approximating an ideal predictable pattern, NA BLE reflects the health status of the circadian rhythm, where irregular light exposure may lead to adverse health outcomes [31,32,33]. A large NA BLE indicates consistent daytime BLE patterns, while minimizing nighttime BLE promotes robust circadian alignment. A small NA BLE encompasses weak daytime BLE, inconsistent daily patterns, and excessive nighttime BLE, all of which can disrupt circadian rhythms. The calculation of NA BLE over a continuous seven-day period provides a reliable measure of the stability of LE patterns, which is important for understanding its effect on lipid metabolism. Furthermore, the circadian robustness and sleep of individuals with a history of COVID-19 were found to be more vulnerable to a small NA BLE [25].

Endogenous rhythms in blood lipids have been documented. Notably, while the peak times (acrophases) of TC, LDL-C, HDL-C, and TG occur in the afternoon (13:00–17:00) under baseline conditions, the TG acrophase shifts dramatically to 03:37 (nocturnal hours) under constant routine conditions [13,34], aligning closely with the melatonin acrophase [13]. Furthermore, blood lipid rhythms are influenced by environmental factors and can be adjusted by shifts in LE and physical activity. Circadian rhythms of blood lipids are known to change with age [35], typically exhibiting a reduced amplitude and a phase advance, consistent with general age-related changes in circadian rhythms [36]. Since LE and physical activity both support large-amplitude circadian rhythms [37], and both often decrease with age, these lifestyle changes may contribute to the observed alterations in blood lipid rhythms. Our study revealed a link between light at night (LAN) and unfavorable alterations in TC and LDL-C concentrations, consistent with recent findings in Chinese adults [16] and an earlier Japanese study that associated LAN with elevated TG and LDL-C in an elderly cohort [2]. REV-ERBs, as a subfamily of nuclear receptors that serve as a crucial component of the core circadian clock gene loop, were identified as a critical regulator of the circadian clock, with significant roles in circadian modulation of lipid metabolism [38]. In Arctic residents, REV-ERBα expression exhibits a seasonal rhythm peaking during the summer and correlates with physical activity, daylight exposure, and population origin [39], suggesting a role in seasonal adaptation and metabolic variation at high latitudes.

As previously described [12], the natural photoperiod duration was significantly modified by artificial LE/BLE. Despite this, average measures of LE/BLE (e.g., MESOR, M10) showed no significant association with blood lipids. In contrast, the timing and amplitude of LE/BLE exhibited distinct associations with specific lipid profiles, which supports the circadian light hypothesis of lipid modulation. Furthermore, recent studies have increasingly emphasized the role of TG, HDL-C, and the TG/HDL-C ratio as independent predictors of cardiovascular risk, particularly in individuals with metabolic disorders [29,30,40]. The observation that these three proxies of cardiometabolic health are most strongly associated with NA BLE further substantiates the importance of circadian light patterns in maintaining acceptable blood lipid profiles.

Our findings regarding the temporal associations between BLE and TG/HDL-C depicted in Figure 2 are in line with our previous research on primary open-angle glaucoma (POAG) [10,41]. We identified specific associations between higher morning TG and lower HDL-C with retinal ganglion cell loss in advanced stages of POAG, where intrinsically photosensitive RGCs are also damaged [42], suggesting compromised light signal perception. Analogous to the natural light deficiency during WS, we hypothesized that reduced BLE would exacerbate the influence of LE timing on lipid metabolism. Consistent with this hypothesis, we observed that while TG concentrations were increased under nocturnal BLE, HDL-C depended on morning BLE, especially during WS. We propose that when natural LE is low (as during WS) or light perception is compromised (as in advanced POAG), the susceptibility of TG and HDL to compromised light hygiene is increased. This vulnerability is further supported by multilinear models demonstrating the predictive power of LE in relation to TG, suggesting that in conditions of weakened melanopsin signaling, even subtle disruptions in LE patterns can significantly impact metabolic parameters.

This study has practical implications: prioritizing earlier daytime LE and minimizing nighttime BLE may potentially improve lipid profiles and cardiometabolic health among individuals experiencing seasonal light variations. This approach may serve as an effective strategy to mitigate the adverse effects of disrupted circadian rhythms on lipid metabolism, particularly in populations at risk.

Strengths: The merit of this study is that data from all participants were collected within the same week in each season in an Arctic region without Daylight Saving Time for over a decade. Studying the same participants across seasons provides robust seasonal comparisons. Furthermore, the melatonin results were obtained from participants with similar scores on the Morningness-Eveningness Questionnaire (MEQ). This homogeneity minimizes chronotype-related bias, though it limits the generalizability of findings to individuals with more extreme chronotypes.

Limitations: The relatively small sample size limits the generalizability of our findings. While the single-point-in-time blood lipid measurements preclude a comprehensive assessment of 24 h lipid dynamics and seasonal variations in their circadian patterns, these measurements are clinically relevant as they reflect the standard morning fasting lipid profiles commonly obtained in medical practice. Future studies should address whether circadian light patterns induce changes in mean lipid values or in the 24 h amplitude and/or phase of lipids.

## 5. Conclusions

This study underscores the primary role of the timing and dynamic range of light, rather than the photoperiod duration or amount of daylight, in predicting blood lipid concentrations among Arctic residents, suggesting that circadian light patterns modulate blood lipids. These findings challenge the traditional focus on photoperiods and highlight the potential for light hygiene interventions to improve cardiometabolic health in high-latitude populations. Future research should prioritize strategies to maintain regular light exposure patterns and good sleep hygiene, thereby mitigating cardiovascular risks and enhancing metabolic health within Arctic communities.

## Figures and Tables

**Figure 1 biology-14-00799-f001:**
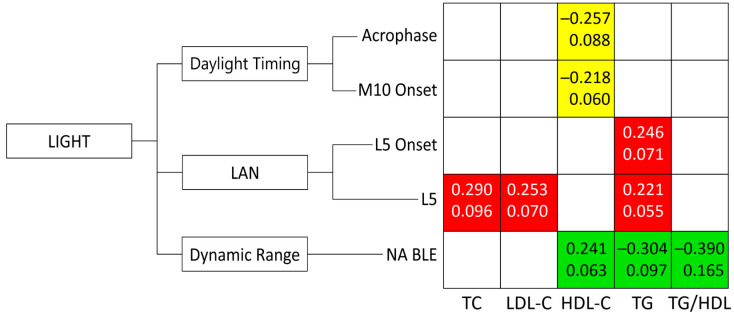
Effect of light exposure on lipid profiles in Arctic residents. This figure illustrates the associations between various aspects of blue light exposure (BLE) and lipid metabolism (Total Cholesterol [TC], LDL-Cholesterol [LDL-C], HDL-Cholesterol [HDL-C], Triglycerides [TG], TG/HDL ratio), adjusted for sex, age, indigeneity, and photoperiod duration. Beta (β)—upper values within each cell indicate the strength and direction of the association. Partial eta-squared (η^2^)—lower values within each cell indicate the proportion of variance in lipids explained by each independent variable, controlling for other predictors in the model. LAN—Light-At-Night, NA BLE—Normalized Amplitude, Amplitude-to-MESOR ratio. Yellow: Represents beneficial associations with daylight patterns: higher HDL-C is linked to earlier timing of BLE onset and daytime BLE. Red: Represents hazardous associations with LAN: Higher TC, LDL-C, and TG are associated with higher nighttime BLE. Green: indicates beneficial associations where a stronger daily light/dark cycle (circadian light hygiene, approximated by the Normalized Amplitude of BLE, NA BLE) is linked to higher HDL-C and lower TG/HDL-C.

**Figure 2 biology-14-00799-f002:**
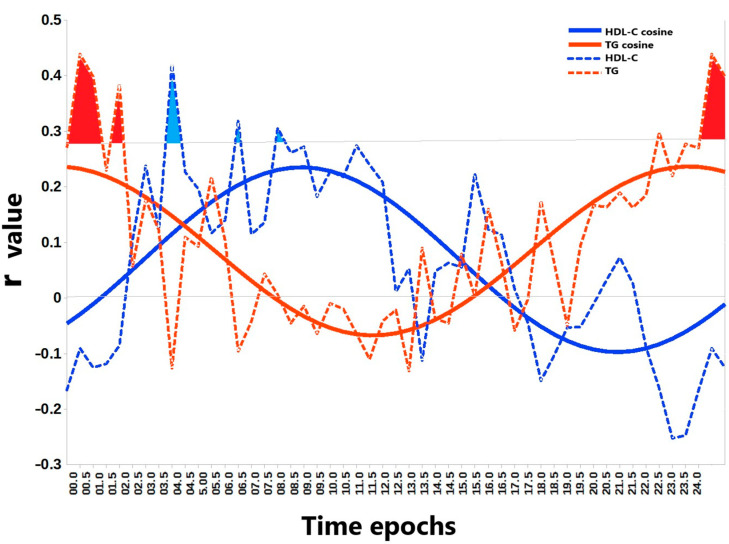
Time-dependent associations between blue light exposure and lipids. This figure represents the correlations between morning lipid values and blue light exposure at various time points across the 24 h day, illustrating the temporal relationship between blue light exposure and two lipid markers: high-density lipoprotein cholesterol (HDL-C) and triglycerides (TG). The plot displays r values (Pearson correlation coefficients) derived from a linear regression of HDL-C (blue curve) and TG (red curve) with blue light exposure within consecutive 30 min time epochs across the 24 h period. The x-axis (abscissa) represents these time epochs, while the y-axis (ordinate) indicates the corresponding r value. The horizontal gray line represents the Benjamini–Hochberg corrected significance threshold (α = 0.1) for multiple comparisons. The dotted curves represent the r values and the solid curves are the corresponding fitted cosinor models. These models significantly reject the null hypothesis of no rhythmicity for both HDL-C (F = 26.9, *p* < 0.00001) and TG (F = 32.1, *p* < 0.00001). Cosinor analysis indicates a morning acrophase (peak) for HDL-C at 08:55 and a near-midnight acrophase for TG at 23:38. Time epochs showing significant correlations after Benjamini–Hochberg correction are shaded in blue (for HDL-C) and red (for TG).

**Table 1 biology-14-00799-t001:** Associations between blood lipids and actigraphy measures of light exposure and melatonin.

Variable	TC	LDL-C	HDL-C	TG	TG/HDL
Photoperiod	0.070	0.033	0.022	−0.003	−0.110
LE MESOR	0.128	0.119	−0.049	0.126	−0.067
LE Amplitude	0.039	0.038	−0.087	0.036	−0.115
LE Acrophase	−0.218	−0.153	**−0.280 ***	−0.051	0.121
BLE MESOR	0.116	0.111	−0.052	0.096	−0.071
BLE Amplitude	0.030	0.039	−0.088	0.019	−0.109
BLE Acrophase	**−0.222**	−0.161	**−0.282 ***	−0.041	0.112
M10 BLE	0.090	0.086	−0.053	0.061	−0.095
M10 Onset BLE	−0.216	−0.121	**−0.268 ***	−0.126	0.007
L5 BLE	**0.312 ***	**0.266**	0.154	**0.234**	0.074
L5 BLE Onset	**0.231**	**0.229**	0.015	**0.255**	0.208
RA BLE	−0.098	−0.074	−0.036	−0.162	−0.154
NA BLE	−0.113	−0.142	0.193	**−0.331 ***	**−0.405 ***
Melatonin acrophase	0.060	0.123	** *−0.282* **	**0.454 ***	**0.510 ***

Note: r values from linear regression of overall data are shown. Significant (*p* < 0.05) associations are in **bold**, borderline significant in ***bold italic***. * Significant after Benjamini–Hochberg’s correction for multiple testing at FDR = 0.1. TC—Total Cholesterol, TG—Triglycerides, HDL-C—High-Density Lipids–Cholesterol, LDL-C—Low-Density Lipids–Cholesterol, MESOR—Midline Estimating Statistic of Rhythm, a rhythm-adjusted mean, M10—10 h of highest values, L5—5 h of lowest values, RA—Relative Amplitude, BLE—Blue Light Exposure, NA—Normalized Amplitude, Amplitude-to-MESOR ratio.

**Table 2 biology-14-00799-t002:** Univariate blood lipid regression models with light exposure predictors and melatonin, adjusted for photoperiod.

Variable	TC		LDL-C		HDL-C		TG		TG/HDL-C	
	β R^2^	*p*	β R^2^	*p*	β R^2^	*p*	β R^2^	*p*	β R^2^	*p*
Timing of Light Exposure										
LE acrophase	---	*---*	---		**−0.280** **0.079**	**0.012**	---	---	---	
BLE acrophase	**−0.221** **0.054**	**0.049**	---	---	**−0.282** **0.080**	**0.012**	---	---	---	
M10 BLE Onset	---	---	---		**−0.274** **0.073**	**0.017**	---	---	---	
L5 BLE Onset	**0.233** **0.059**	**0.039**	**0.230** **0.054**	**0.042**	---	---	**0.255** **0.065**	**0.023**	** *0.206* ** ** *0.055* **	** *0.067* **
Nocturnal Blue Light Exposure										
L5 BLE	**0.309** **0.098**	**0.006**	**0.267** **0.071**	**0.026**	---	---	**0.240** **0.056**	**0.036**	---	--- **0.098**
Dynamic Range of Light Exposure										
NA BLE	---	---	**0.002**	**0.345** **0.114**	** *0.197* ** ** *0.037* **	** *0.089* **	**0.345** **0.114**	**0.002**	**0.399** **0.164**	**<0.001**
Melatonin										
Melatonin acrophase	---	---	---	---	**−*0.297*** ** *0.087* **	** *0.057* **	**0.473** **0.217**	**0.002**	**0.503** **0.250**	**<0.001**

Note: Only significant or borderline significant predictors are listed. TC—Total Cholesterol, TG—Triglycerides, HDL-C—High-Density Lipids-Cholesterol, LDL-C—Low-Density Lipids–Cholesterol, LE—Light Exposure, BLE—Blue Light Exposure, MESOR—Midline Estimating Statistic of Rhythm, a rhythm-adjusted mean, M10—BLE during the 10 most exposed hours, L5 BLE—BLE during the 5 least exposed hours, NA BLE—Normalized Amplitude of BLE. Significant (*p* < 0.05) associations are in **bold**. Borderline significant are in ***bold italic***.

**Table 3 biology-14-00799-t003:** Mean values of light exposure per season.

Variable	Winter Solstice (WS)	Spring Equinox (SE)	Summer Solstice (SS)
LE MESOR, lux	23.68 ± 13.92	74.01 ± 46.88 *	213.67 ± 234.66 **/**
LE Amplitude, lux	25.75 ± 14.78	103.93 ± 67.53 *	276.70 ± 340.14 **/**
LE Acrophase, hh:mm	14:14 ± 1:15	12:57 ± 0:58 **	14:12 ± 1:36 ^ns^/**
BLE MESOR, μw/cm^2^	2.64 ± 1.61	11.11 ± 7.55 *	36.66 ± 43.81 **/**
BLE Amplitude, μw/cm^2^	2.93 ± 1.74	16.12 ± 11.19 *	49.91 ± 65.04 **/**
BLE Acrophase, hh:mm	14:12 ± 1:12	12:52 ± 0:57 **	14:06 ± 1:33 ^ns/^**
M10 BLE, μw/cm^2^	4.99 ± 2.92	23.48 ± 15.35 *	67.62 ± 78.70 **/**
M10 Onset BLE, hh:mm	8:54 ± 1:06	7:52 ± 0:49 **	8:20 ± 1:08 */*
L5 BLE, μw/cm^2^	0.128 ± 0.213	0.077 ± 0.019 ^ns^	0.816 ± 2.871 */*
L5 BLE Onset, hh:mm	2:04 ± 1:31	1:20 ± 1:49 *	1:42 ± 1:19 ^ns/ns^
RA BLE, a.u.	0.934 ± 0.110	0.983 ± 0.026 **	0.960 ± 0.059 ^ns/ns^
NA BLE, a.u.	1.12 ± 0.20	1.43 ± 0.16 **	1.25 ± 0.29 */**

Mean values ± Standard Deviation (SD) are indicated. MESOR—Midline Estimating Statistic of Rhythm, a rhythm-adjusted mean, M10—10 h of highest values, L5—5 h of lowest values, BLE—Blue Light Exposure, NA—Normalized Amplitude, Amplitude-to-MESOR ratio. ** *p* < 0.01; * *p* < 0.05: SE—for difference vs. WS: SS—for differences vs. WS/SE. ns, not significant.

**Table 4 biology-14-00799-t004:** Demographic predictors of blood lipids in fully adjusted (indigeneity, photoperiod duration, sex, age) multiple regression model.

	TC		LDL-C		HDL-C		TG		TG/HDL-C	
	β	*p*-Value	β	*p*-Value	β	*p*-Value	β	*p*-Value	β	*p*-Value
Age	**0.261**	**0.023**	** *0.244* **	** *0.053* **	−0.017	0.873	0.160	0.164	0.138	0.212
Sex (m)	−0.091	0.447	0.070	0.056	−**0.412**	**<0.001**	0.127	0.295	**0.238**	**0.044**
Indigeneity (NN)	0.004	0.969	0.065	0.578	−0.167	0.124	**0.333**	**0.003**	**0.288**	**0.010**
Photo-period	0.061	0.565	0.021	0.847	0.034	0.727	−0.029	0.785	−0.132	0.220

Note: Significant (*p* < 0.05) associations are in **bold**. Borderline significant are in ***italic bold***. TC—Total Cholesterol, TG—Triglycerides, LDL-C—Low-Density Lipid Cholesterol, HDL-C—High-Density Lipid Cholesterol, TG—Triglycerides, MESOR—Midline Estimating Statistic of Rhythm, a rhythm-adjusted mean.

## Data Availability

The data presented in this study are available on reasonable request from the corresponding author. The data are not publicly available due to privacy.

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
