# Peer review of "Timing and Amplitude of Light Exposure, Not Photoperiod, Predict Blood Lipids in Arctic Residents: A Circadian Light Hypothesis"

_biology, 2025, doi:10.3390/biology14070799_

Round 1

Reviewer 1 Report

Comments and Suggestions for Authors

Dear Authors,

Thank you for interesting article focusing on lipid levels in residents living in extreme photoperiods. The data from real-life condition studies are highly relevant for understanding of human physiology, and for validation of laboratory studies.

I have consecutive  comments and questions:

Materials and methods:

  1. Page 3, L 140: In the text you have reported that blood was collected by venipuncture between 8:00 and 9:00 a.m.. The blood was collected just once and only in this time window or was collected repeatedly in specific intervals? In the Figure 2 you have lipids in 0,5 h intervals. Specify the information about the blood collecting.

  1. Do you have any data about diet (particularly the fat proportion) in native and non native residents or in men and women especially before blood collecting? Had the residents some dietary recommendations? Add the information.

  1. From the text it is not clear if the blue light exposure (BLE) was planned and included in the experimental design in some way, or BLE was just measured during quotidian life of residents (natural BLE + artificial BLE in work and home). Clarify it in the text. If it was planned and included, add this information to the „Materials and Methods“.

Results:

  1. Page 5, L230-236: The results about melatonin are presented in the text. For more complex information add the relevant statistics to table (table 1 or separate table).

  1. Page 5, L244-248: I miss the table with results from multi-factorial analysis for age, sex and native status (as presented in the discussion (L303)). Please, add this table.

  1. Page 6, table1 (L251-255) RA BLE is not characterized

References:

  1. I found 12 self-citations out of a total 42 references. Although, the topic of your work is specific, some references dealt with a broader thematic scope, for example 23 (Danilenko et al., 2022), 36 (Gubin et al., 2016) and 37 (Weinert and Gubin, 2022). These references should be replaced, or at least they should be supplemented with references from other authors/research group.

Author Response

Dear Authors,

Thank you for interesting article focusing on lipid levels in residents living in extreme photoperiods. The data from real-life condition studies are highly relevant for understanding of human physiology, and for validation of laboratory studies.

I have consecutive comments and questions:

R1: We are grateful to reviewer for thorough evaluation of our work and raising important questions.

Q2. Materials and methods:

Page 3, L 140: In the text you have reported that blood was collected by venipuncture between 8:00 and 9:00 a.m.. The blood was collected just once and only in this time window or was collected repeatedly in specific intervals? In the Figure 2 you have lipids in 0,5 h intervals. Specify the information about the blood collecting.

A2: Thank you for pointing out the potential ambiguity. Blood samples were collected only once per season, in the morning between 8:00 and 9:00 a.m. as detailed in the Materials and Methods section on page 3, line 140. The data presented in Figure 2 do not represent repeated blood draws. Instead, Figure 2 displays the results of a regression analysis. Specifically, we used a cosinor modeling approach to analyze the relationship between morning lipid levels (HDL-C and TG) and light exposure (as the independent variable) across consecutive 30-minute time epochs throughout the 24-hour day. This approach allowed us to assess the rhythmicity of this relationship, particularly during the winter solstice, which was the most challenging season for light exposure. The r-values represent the correlation between morning lipid values and light exposure at the different specified time epochs. In essence, the analysis provides a linear, time-series-based alternative to grouping analysis, showing how individual morning lipid values relate to light exposure at various time points, rather than comparing light exposure across patient groups with defined threshold levels.

We now added the following clarifying note to the figure legend: “This figure represents the correlations between morning lipid values and blue light exposure at various time points across the 24-hour day, illustrating the temporal relationship between blue light exposure and two lipid markers: high-density lipoprotein cholesterol (HDL-C) and triglycerides (TG).

Q3 Do you have any data about diet (particularly the fat proportion) in native and non native residents or in men and women especially before blood collecting? Had the residents some dietary recommendations? Add the information.

A3: No specific dietary recommendations were provided to the participants beyond the standard instruction to fast prior to blood collection for lipid analysis. Participants were otherwise encouraged to maintain their usual daily activities, including their regular dietary habits. While we did not collect detailed dietary data, we are not aware of any significant differences in fat consumption between native and non-native residents in this study population. All participants, including the native residents, were integrated into the local society and followed a standard local diet.

The following clarifying text was now added to the Methods section: “Following the conclusion of actigraphy monitoring, participants presented at the clinic in the morning (between 8:00 and 9:00 a.m.), after a 12-h period of fasting, for blood sampling. Participants were encouraged to maintain their usual daily activities and dietary habits.”

Q4 From the text it is not clear if the blue light exposure (BLE) was planned and included in the experimental design in some way, or BLE was just measured during quotidian life of residents (natural BLE + artificial BLE in work and home). Clarify it in the text. If it was planned and included, add this information to the „Materials and Methods“.

A4. The blue light exposure (BLE) in this field study was not planned or manipulated. Instead, BLE was measured during the participants’ normal daily lives, encompassing both natural outdoor light and artificial indoor light from their homes and workplaces. This approach, reflecting real-world light exposure patterns, was a crucial element of our two-phase study design, as described in our prior publications [cited in Methods]. The first phase, presented in this manuscript, focuses on characterizing baseline light exposure and its associations with various physiological parameters. The second, ongoing phase will involve an intervention using human-centric lighting. The timing and nuances of that intervention will be informed by the real-life light exposure data collected during the first phase, across different seasons. This naturalistic approach is essential for understanding the impact of typical light patterns on health and well-being.

We now added the following clarification to the Methods: “LE and BLE was assessed in a natural setting, reflecting participants’ real-world light exposure to inform the human-centric lighting intervention in the ongoing second phase of the study.

Q5 Results: Page 5, L230-236: The results about melatonin are presented in the text. For more complex information add the relevant statistics to table (table 1 or separate table).

A5: We appreciate the suggestion from this reviewer. We now added corresponding results and relevant statistics on melatonin to Tables 1 and 2.

Q6 Page 5, L244-248: I miss the table with results from multi-factorial analysis for age, sex and native status (as presented in the discussion (L303)). Please, add this table.

A6: We are grateful to the Reviewer for this suggestion. A new Table 4 (Table 4. Demographic Predictors of Blood Lipids in Fully Adjusted (Indigeneity, Photoperiod Duration, Sex, Age) Multiple Regression Model.) was now added.

Q7 Page 6, table1 (L251-255) RA BLE is not characterized

A7: Thank you for noting. Done.

Q8 References:

I found 12 self-citations out of a total 42 references. Although, the topic of your work is specific, some references dealt with a broader thematic scope, for example 23 (Danilenko et al., 2022), 36 (Gubin et al., 2016) and 37 (Weinert and Gubin, 2022). These references should be replaced, or at least they should be supplemented with references from other authors/research group.

A8: We have replaced references 23, 36, and 37 with citations from other groups, as recommended. We have retained only self-citations directly supporting the methodology, establishing the study context, and explaining the hypothesis/aims, essential for clarity and accuracy in conveying the study’s findings and rationale.

Reviewer 2 Report

Comments and Suggestions for Authors

This is an unusually novel and sophisticated study describing important findings for both basic research on biological clocks, and for biomedical relevance. The database is unique, the analyses are thorough and the findings are significant and relevant to a large proportion of the human population if the findings generalize to less extreme latitudes.  The methods capture the complexity of interactions between photoperiod, spectral quality of light, time of day and human circadian clock dynamics. This is especially intriguing because some arctic organisms appear to show only weak endogenous circadian clock regulation of behavior. Although the tables and text describing many (important) statistical associations among acronyms is somewhat confusing, Figures 1 and 2 provide excellent graphical representations of the patterns in the data.

I only have a few very minor suggestions:

Materials and Methods, line 119: Note that the three towns are in Russia. 

Discussion, lines 329-330: should "melatonin phase" be "melatonin acrophase"? 

Discussion, line 330: Delete "research demonstrates"?

Lines 399-400: Supplementary Materials: Include these as another figure in the text? It's is only a small additional amount of data but it is relevant and, in my opinion, worth including in the main text. It's more trouble (to readers) than it's worth to bury one small table in a supplemental file.  

Author Response

This is an unusually novel and sophisticated study describing important findings for both basic research on biological clocks, and for biomedical relevance. The database is unique, the analyses are thorough and the findings are significant and relevant to a large proportion of the human population if the findings generalize to less extreme latitudes. The methods capture the complexity of interactions between photoperiod, spectral quality of light, time of day and human circadian clock dynamics. This is especially intriguing because some arctic organisms appear to show only weak endogenous circadian clock regulation of behavior. Although the tables and text describing many (important) statistical associations among acronyms is somewhat confusing, Figures 1 and 2 provide excellent graphical representations of the patterns in the data.

We appreciate the Reviewer’s thorough evaluation of our work and are grateful for the encouraging feedback.

I only have a few very minor suggestions:

Comments:

Q1: Materials and Methods, line 119: Note that the three towns are in Russia.

A1: Done.

Q2: Discussion, lines 329-330: should "melatonin phase" be "melatonin acrophase"?

A2: Changed text to acrophase.

Q3: Discussion, line 330: Delete "research demonstrates"?.

A3: Done.

Q4: Lines 399-400: Supplementary Materials: Include these as another figure in the text? It's is only a small additional amount of data but it is relevant and, in my opinion, worth including in the main text. It's more trouble (to readers) than it's worth to bury one small table in a supplemental file.

A4: We are grateful to the Reviewer for this suggestion. We now moved Supplemental Table to the main text. It is now Table 3.